# REASONEDIT: TOWARDS REASONING-ENHANCED IMAGE EDITING MODELS

## ABSTRACT

Recent advances in image editing models have shown remarkable progress. A common architectural design couples a multimodal large language model (MLLM) encoder with a diffusion decoder, as seen in systems such as Step1X-Edit and Qwen-Image-Edit, where the MLLM encodes both the reference image and the instruction but remains frozen during training. In this work, we demonstrate that unlocking the reasoning capabilities of MLLM can further push the boundaries of editing models. Specifically, we explore two reasoning mechanisms, thinking and reflection, which enhance instruction understanding and editing accuracy. Based on that, our proposed framework enables image editing in a thinking–editing–reflection loop: the thinking mechanism leverages the world knowledge of MLLM to interpret abstract instructions, while the reflection reviews editing results and automatically corrects unintended manipulations. Extensive experiments demonstrate that our reasoning approach achieves significant performance gains over the base model, with improvements of ImgEdit (+0.7%), GEdit (+1.4%), and Kris (2.5%), and also outperforms previous open-source methods on both Kris and ImgEdit. **Code will be open-sourced.**

## 1 INTRODUCTION

Image editing with diffusion models has witnessed rapid progress, moving from early mask-based approaches such as BrushNet (Ju et al., 2024) and PowerPaint (Zhuang et al., 2024), to instruction-driven systems like InstructPix2Pix (Brooks et al., 2022) and OmniGen (Xiao et al., 2025), and more recently to multimodal frameworks that integrate an MLLM encoder with a diffusion decoder, like Step1X-Edit (Liu et al., 2025) and Qwen-Image-Edit (Wu et al., 2025a). These advances have substantially improved controllability and usability, enabling more diverse and flexible image editing. However, state-of-the-art instruction-based methods still face challenges in generalizing instructions, as most models keep MLLM encoders frozen during training. As a result, current models exhibit limited visual reasoning capabilities, which restricts their ability to handle complex or abstract instructions. More importantly, such limitations prevent them from benefiting fully from test-time scaling, a paradigm that has driven significant improvements in language models.

Turning to the visual reasoning domain, recent advances have explored reasoning-enhanced visual generation through unified understanding and generation (Deng et al., 2025), reflection-based refinement (Wu et al., 2025c; Li et al., 2025), and chain-of-thought modeling (Zhang et al., 2025a; Wang et al., 2025b; Huang et al., 2025). These studies highlight the potential of reasoning for controllable and efficient generation. For instance, BAGEL (Deng et al., 2025) introduces a thinking mode that leverages the world knowledge of MLLMs to interpret abstract instructions in image generation and editing, while OmniGen2 (Wu et al., 2025c) integrates reflection capabilities of MLLMs into image generation. Despite these advances, most existing efforts remain centered on image generation, leaving the application of reasoning to image editing largely underexplored. A key underlying challenge lies in the substantial hallucinations of MLLMs during paired image understanding, particularly in capturing the differences between reference and edited results (Fu et al., 2024; Wang et al., 2025a) and in generating appropriate refined instructions for subsequent editing.

To this end, we propose ReasonEdit, a fundamental editing model with two reasoning capbilities: **thinking** and **reflection**. The former primarily transforms ambiguous, colloquial, and informal editing instructions into clear, standardized, and actionable directives by constructing Thinking Pairs,

Figure 1: Illustration of the ReasonEdit's reasoning capabilities. The thinking module illustrates how a model decomposes abstract instructions into clear, actionable commands. The reflection pipeline, conversely, showcases the model's ability to perform an iterative self-correction loop, refining an intermediate generated image to achieve a more accurate final result.

which are structured as abstract-to-concrete instruction pairs. The latter is designed to perform iterative self-correction and refinement during the editing process by restructuring the paired image understanding as multiple cascaded single-image understanding tasks. We achieve this by constructing Reflection Triples that form an iterative cycle: <original image, editing instructions, edited image, reflection instructions, reflection-corrected image>. To train these image editing reasoning capabilities, our network architecture integrates a MLLM as the Reasoner and a DiT as the Generator. We employ a multi-stage training strategy: initially, the model is trained independently on image editing and thinking tasks, followed by a joint training phase. This progressive approach simplifies the learning objectives at each stage, leading to smoother convergence and a more effective, gradual acquisition of both editing and reasoning capabilities.

Our contributions can be summarized as follows:

- A reasoning-enhanced editing model that natively supports a thinking–editing–reflection workflow. Thinking mode allows parsing original instructions leveraging the world knowledge of MLLMs, enabling the model to tackle more complex editing tasks. Reflection mode enables iterative refinement by reviewing and correcting the results of previous edits.

- A comprehensive data construction pipeline consisting of <original image, editing instructions, edited image, reflection instructions, reflection-corrected image>, which supports end-to-end training of the thinking–editing–reflection loop.

## 2 RELATED WORK

### 2.1 IMAGE EDITING MODELS

Diffusion models have demonstrated remarkable progress in generative modeling, particularly for producing high-fidelity and diverse image editing results. Early approaches, such as BrushNet (Ju et al., 2024), BrushEdit (Li et al., 2024), PowerPaint (Zhuang et al., 2024), and FLUX.1-Fill-dev (Black Forest Labs, 2024b), typically employ an edit-area mask together with textual instructions to achieve localized and high-quality edits. Beyond mask-based control, recent works have further explored enhancing editing controllability by incorporating multiple visual conditions. For instance, OminiControl (Tan et al., 2025), ACE (Han et al., 2025), and ACE++ (Mao et al., 2025) unify diverse conditional signals such as depth maps and keypoints within a single model, thereby enabling more flexible and versatile editing capabilities.

While visual conditions offer precise control, they also raise the usage threshold. In contrast, instruction-based models aim to enable editing through natural language, but often struggle to align semantic understanding with fine-grained manipulation. Pioneering efforts such as Instruct-Pix2Pix (Brooks et al., 2022), MagicBrush (Zhang et al., 2023), UltraEdit (Zhao et al., 2024), AnyEdit (Yu et al., 2025), and OmniGen (Xiao et al., 2025) construct large-scale instruction–image pairs to support purely instruction-driven editing, yet still face challenges in fidelity and quality. Recent approaches address this challenge by leveraging priors from advanced text-to-image mod-

els (Black Forest Labs, 2024a;c; Cai et al., 2025), as in ICEdit (Zhang et al., 2025b), Hidream-E1 (HiDream-ai, 2025), and FLUX.1-Kontext-dev (BlackForestLabs et al., 2025). Another line of work integrates multimodal large language model encoders with diffusion decoders, such as Qwen2VL-Flux (Lu, 2024), MetaQueries (Pan et al., 2025), BLIP3-o (Chen et al., 2025), UniWorld-v1 (Lin et al., 2025), Step1X-Edit (Liu et al., 2025), and Qwen-Image-Edit (Wu et al., 2025a).

Although existing models have achieved notable progress in instruction-based editing, their reliance on frozen MLLM encoders limits performance on complex or abstract instructions. Motivated by this, ReasonEdit unlocks the reasoning capability of MLLMs through joint optimization with the diffusion decoder, thereby improving semantic understanding and extending the boundaries of controllable image editing.

## 2.2 REASONING-ENHANCED VISUAL GENERATION MODELS

The test-time scaling paradigm has rapidly extended from language to multimodal domains, giving rise to several reasoning-enhanced visual generation models. ThinkDiff (Mi et al., 2025) introduces multimodal in-context reasoning into diffusion models via a "think-then-diffuse" inference scheme, while BAGEL (Deng et al., 2025) enables a thinking mode by jointly training visual understanding and generation tasks. In addition to pre-thinking mechanisms before generation, some works explore reflection strategies to refine outputs, such as OmniGen2 (Wu et al., 2025c) and Reflect-DiT (Li et al., 2025). Others, including Image-CoT (Zhang et al., 2025a), MINT (Wang et al., 2025b), and IRG (Huang et al., 2025), employ multimodal chain-of-thought reasoning to guide the generation process. Beyond text-to-image, GoT (Fang et al., 2025) integrates reasoning with diffusion models using large-scale reasoning-chain data for controllable generation and editing. Uni-CoT (Qin et al., 2025) further decomposes multimodal chain-of-thought learning into macro- and micro-level components with auxiliary tasks, enabling efficient training for complex reasoning.

Compared with the above approaches, ReasonEdit focuses more on exploring thinking and reflection mechanisms for editing tasks, enhancing instruction understanding and editing accuracy. While Uni-CoT (Qin et al., 2025) is a concurrent work, our method adopts different base models, training data composition, and training paradigm.

## 3 METHOD

This section introduces the training data construction and the training of our model. We first elaborate on the construction of our edit reasoning data. Following this, we describe the training of the proposed REASONEDIT, in which we present the model design, the multi-stage training strategy. Finally, and a thorough account of the specific training details, ensuring clarity of the our method.

### 3.1 DATA CONSTRUCTION

To facilitate supervised fine-tuning of our reasoning model, we have developed two distinct datasets: thinking and reflection. The former consists of abstract instruction-clear multi-step decomposition instruction pairs, while the latter includes triples that encompass multiple cascaded single-image understanding tasks.

**Thinking Pairs** consist of abstract-to-concrete instruction pairs. Each pair links an abstract instruction, which captures a user's original request in ambiguous, colloquial, or informal language, with its corresponding set of concrete, actionable commands. The concrete counterpart translates the initial user intent into one or more precise, standardized, and executable directives. For instance, the abstract entry "symptoms of potassium deficiency in leaves" is paired with the concrete command "Render the leaves yellow and desiccate the leaf tips." For more complex requests, this structure facilitates a logical decomposition into a single, cascaded sequence of directives. As an illustration, a multifaceted request like "Make the image more dramatic with a vintage feel" is deconstructed into a single, composite instruction: "Increase the image contrast. Apply a sepia tone filter. Add a subtle vignette effect".

To construct the Thinking Pairs dataset, we devised a three-step process combining categorization, annotation, and review, leveraging different advanced Vision-Language Models (VLMs) as annotators. First, we classified a large pool of raw instructions as either already clear or as abstract

and complex. Then, in a two-way annotation process, we generated the corresponding abstract instructions for the clear commands and decomposed the complex instructions into clear, actionable sub-directives. Finally, a rigorous review ensured that each pair met our specific abstract-to-concrete requirements. We also supplemented the dataset with a small number of simple instructions that did not require rewriting. This ensures our model can learn to handle both complex requests by decomposing them and simple requests by outputting them directly.

The Thinking Pairs dataset is built from an initial 500k image-instruction pair pool, which we categorize into 112k complex and 388k simple instructions. After annotating the entire set, a rigorous review process selects 150k high-quality abstract-to-concrete pairs. Specifically, 62k of these pairs come from simplifying complex instructions, and 88k are created by adding an abstract layer to simple ones. To ensure versatility, we also include 50k pairs of simple, unedited instructions. The final dataset totals 200k carefully curated pairs.

**Reflection Triples** is constructed from a large collection of existing image-editing pairs, designed to facilitate a model's multi-step, cascaded reasoning capabilities. Each core triple consists of an <Input Image>, a <Generated Image>, and a <Target Image>. This structure models a chained editing process: the <Generated Image> represents an intermediate output from an initial edit on the <Input Image>, providing the crucial context for a multi-round, single-image reflection process through which the model evaluates the generated output and performs subsequent adjustments to produce the refined <Target Image>. In some instances, the generated image and the target image are identical if no further edits are required.

To mitigate hallucination issues inherent in single-pass, dual-image evaluation, we designed a multi-round, single-image reflection pipeline to enable robust reflection. This process begins by generating a target image description based on the input image and instruction, which acts as a faithful and concise blueprint for the intended outcome. A quantitative evaluation then provides a consistency score and rationale, with metrics designed to assess the presence of conflicts, omissions, and hallucinations. This comprehensive evaluation serves as a strong prior for the final reflection and decision-making phase, where the model assesses the edit's success using the original image, the generated image, and the original instruction as a basis. The process yields one of three conclusions: **Successful Edit:** Indicated by reasoning content and a <#Success> tag, signifying consistency between the generated and target images. **Refinable Edit:** For edits that are not fully successful but allow for refinement, reasoning content and a <#Reflection> tag are returned, along with a secondary editing instruction based on the generated image. **Failed Edit:** If an edit fails due to irrecoverable flaws, reasoning content and a <#Failed> tag are provided.

The Reflection Triples is constructed from an initial pool of 500k image-editing pairs. To diversify the modalities of intermediate images, we generate an additional 500k images using four mainstream editing methods (OpenAI, 2025; Shi et al., 2024; Liu et al., 2025; BlackForestLabs et al., 2025). We then apply our previously described reflection pipeline, utilizing an advanced VLM to automate the process. After a final, rigorous manual screening, the curation yields 180k valid data pairs, with an approximate ratio of 3:1:1 for success, reflection, and failed examples.

## 3.2 TRAINING

Based on the reasoning-enhanced dataset, we utilize a multi-stage training strategy to effectively integrate reasoning and image editing capabilities into a single unified model.

### 3.2.1 MODEL DESIGN

As shown in Fig. 2, our model integrates an MLLM as the Reasoner and a DiT (Peebles & Xie, 2023) as a Generator. Specifically, we directly adopt Step1X-Edit (Liu et al., 2025) as our base architecture, which employs Qwen2.5VL 7B Instruct (Bai et al., 2025) for text embedding and a 12B DiT as its diffusion head. In contrast to the original Step1X-Edit, we enhance the MLLM and diffusion transformer with Thinking and Reflection capabilities on image editing. This is achieved through a multi-stage training strategy and subsequent fine-tuning on our reasoning-enhanced dataset, thereby progressively refining the model's performance. It is important to note that, while Step1X-Edit serves as our chosen implementation, our proposed method is broadly applicable across various image editing approaches.

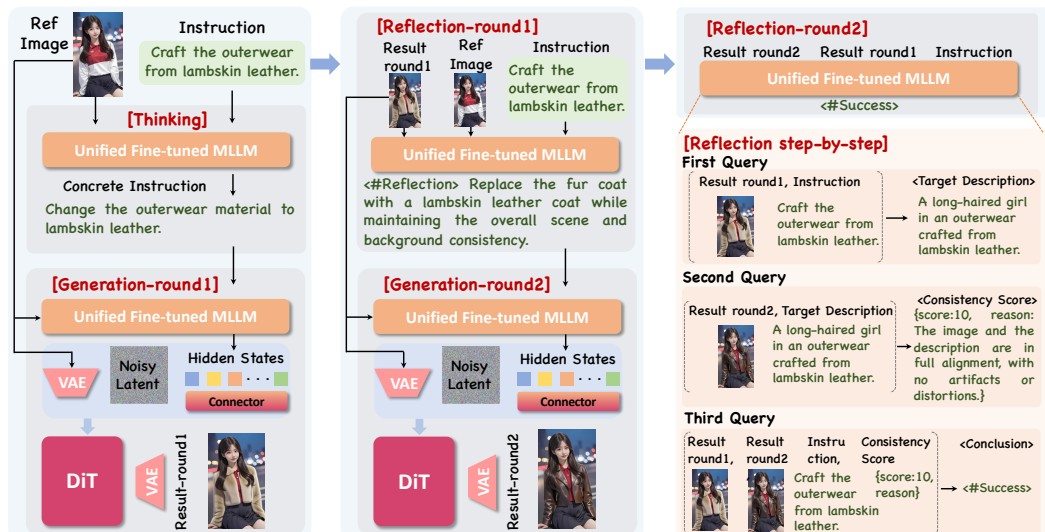

Figure 2: The model architecture and inference pipeline of REASONEDIT are structured around two core components: the Thinking and Reflection processes which serve as the Reasoner, and a DiT acting as the Generator. These Reasoner and Generator modules undergo a multi-stage training process. During inference, they operate in an interleaved and sequential manner, progressively yielding more precise image editing results through their integrated Thinking and Reflection capabilities.

### 3.2.2 MULTI-STAGE TRAINING

Prior work highlights that such reconciliation often necessitates dedicated architectural advancements, for instance, in vision encoders (Ma et al., 2025; Wu et al., 2025b; Qu et al., 2025), to mitigate conflicts during early joint training on both understanding and generation. To address these complex dynamics and effectively integrate enhanced reasoning with generative processes, we adopt a multi-stage training strategy. This progressive approach decomposes the intricate joint optimization into simpler, focused tasks: initially cultivating the MLLM's explicit Thinking and Reflection, subsequently adapting the Generator (DiT) to these refined MLLM on image editing, and culminating in a comprehensive joint fine-tuning of both components to achieve superior overall performance.

REASONING LEARNING STAGE. This initial stage is dedicated to cultivating the MLLM's explicit Thinking and Reflection capabilities tailored for image editing tasks. To efficiently adapt the model while mitigating catastrophic forgetting of its foundational knowledge, and to isolate reasoning training, we employ Low-Rank Adaptation (LoRA) (Hu et al., 2022) on the linear layers in attention modules. During this phase, the DiT remains frozen. Training is conducted on the constructed Thinking Pairs and Reflection Triples datasets ($cf.$ Sec. 3.1), optimizing with a standard Next Token Prediction (NTP) loss,

$$\mathcal{L}_{\text{NTP}} = \mathbb{E}_{t_i} \left[ -\sum_{k=1}^{L} \log p_\theta(t_k|t_1, t_2, \cdots, t_{k-1}) \right] \tag{1}$$

where $t_k$ represents the $k$-th token in a sequence of length $L$, and $p_\theta$ is the probability predicted by the MLLM parameterized by $\theta$.

EDIT LEARNING STAGE. Following the dedicated tuning of the MLLM's reasoning abilities, this stage focuses on adapting the Generator, specifically the DiT model. To leverage the MLLM's refined contextual understanding without interference, its parameters are kept frozen throughout this phase. The DiT is trained using a flow matching loss (Lipman et al., 2023), with a dual objective that encompasses both text-to-image (T2I) generation and direct image editing tasks. Including T2I data is crucial; their significantly larger scale and broader domain coverage are instrumental in enriching the model's general generative knowledge, which in turn substantially improves its proficiency in

diverse editing scenarios. The flow matching loss is formulated as,

$$\mathcal{L}_{\text{FM}} = \mathbb{E}_{t \sim U(0,1), x_0 \sim \mathcal{D}, x_1 \sim \mathcal{N}(0,I), c} \| u_t(x|c) - v_t(x|x_0, c) \|_2^2 \tag{2}$$

where $t$ is uniformly sampled from $[0, 1]$, $x_0$ is a data point from the dataset $\mathcal{D}$, $x_1$ is standard Gaussian noise, and $c$ represents the conditioning information (e.g., a text or a reference image). The DiT model $u_t$ is trained to predict the target vector field $x_1 - x_0$ at the interpolated point $x_t = (1 - t)x_0 + tx_1$.

UNIFIED TUNING STAGE. After the preceding stages, this final stage unifies and jointly fine-tunes both the MLLM and the DiT. This comprehensive joint optimization is crucial for ensuring that the understanding and generative processes seamlessly complement each other. The joint training loss for this stage is formulated as,

$$\mathcal{L}_{\text{joint}} = \mathcal{L}_{\text{FM}} + \omega_{\text{NTP}} \cdot \mathcal{L}_{\text{NTP}} \tag{3}$$

### 3.2.3 TRAINING DETAILS

We utilized Step1X-Edit V1.1 (StepFun AI, 2025) as our pretrained model. The first reasoning learning stage involved training the MLLM on 32 H800 GPUs (4 nodes, 8 GPUs/node) for 16 hours, completing 50,000 steps with an initial learning rate of $1 \times 10^{-4}$. The second edit learning stage scaled to 128 GPUs (16 nodes, 8 GPUs/node), training for 38.9 hours and 28,000 steps at a learning rate of $1 \times 10^{-5}$. Over 38.9 hours and 28,000 steps, with a learning rate of $1 \times 10^{-5}$, the DiT was trained using 14.4 million in-house T2I samples and 2.4 million in-house image editing samples. The final stage consisted of 20 hours of training, completing 12,000 steps with a learning rate of $6 \times 10^{-6}$ and the NTP loss weight $\omega_{\text{NTP}}$ of 0.1. Similar to BAGEL (Deng et al., 2025) and Mogao (Liao et al., 2025), during this stage, FlexAttention (Dong et al., 2024) and a packed data format (Dehghani et al., 2023) were utilized to support efficient hybrid training for both understanding and generation tasks, especially on the Reflection Triples. To optimize training performance and scalability, distributed training employed several parallelization strategies. Specifically, the MLLM and the Connector utilized *sequence parallelism* and *DeepSpeed Ulysses* (Jacobs et al., 2024). For the DiT, both *tensor parallelism* and *sequence parallelism* were applied, enabling effective scaling across multiple nodes and GPUs and accelerating the training process.

## 4 EXPERIMENTS

### 4.1 EXPERIMENTAL SETTINGS

**Benckmark.** We conduct our experiments on three widely-used benchmarks: GEdit-Bench (Liu et al., 2025) and ImgEdit-Bench (Ye et al., 2025) for evaluating broad and comprehensive foundational image editing capabilities, and KRIS-Bench (Wu et al., 2025d) for assessing a model's advanced reasoning skills and ability to interpret abstract instructions. These benchmarks collectively enable a thorough evaluation of our model's performance, ranging from foundational editing tasks to complex, abstract reasoning challenges.

**Metrics.** For the GEdit-Bench, we evaluate performance using three metrics-Semantic Consistency (SQ), Perceptual Quality (PQ), and an Overall Score (O)-which are automatically assessed by VIEScore (Ku et al., 2023) using GPT-4.1. On the ImgEdit-Bench, we use GPT-4.1 to assign 1-5 ratings across three dimensions-instruction adherence, image-editing quality, and detail preservation-where the final score for the latter two is capped by instruction adherence. On the KRIS-Bench, we use GPT-4o to assign 1-5 ratings across four dimensions-Visual Consistency, Visual Quality, Instruction Following, and the novel Knowledge Plausibility, which assesses consistency with real-world knowledge.

### 4.2 EXPERIMENTAL RESULTS

Quantitative results are first reported on GEdit-Bench and ImgEdit-Bench to assess foundational editing capabilities ($cf$. Sec. 4.2.1), with the evaluation then shifting to the more complex KRIS-Bench for an assessment of abstract reasoning skills ($cf$. Sec. 4.2.2).

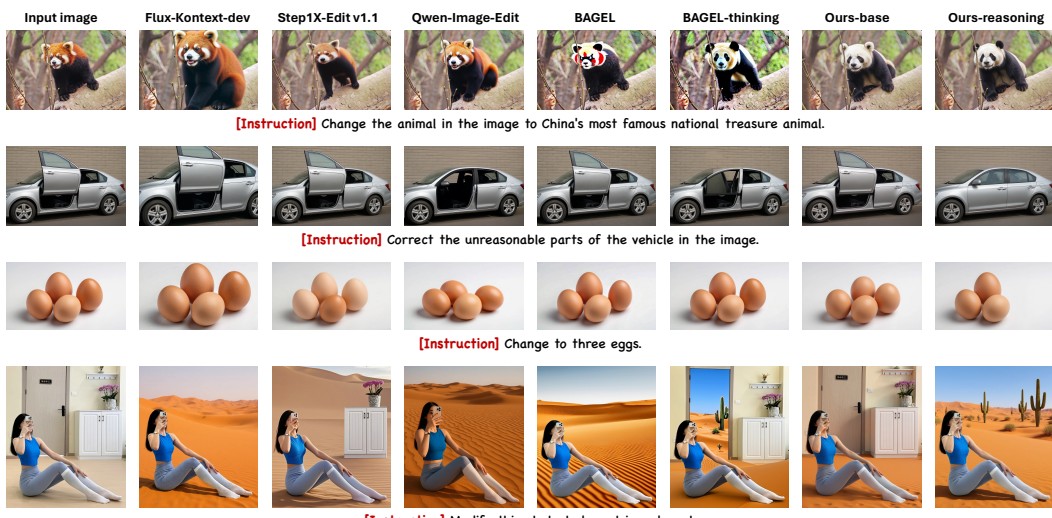

Figure 3: Comprehensive qualitative evaluation of leading image editing models. The results demonstrate that our proposed approach, which incorporates thinking and reflection mechanisms, significantly outperforms the editing model.

Table 1: Comprehensive quantitative evaluation of leading image editing models. Our approach achieves state-of-the-art performance among open-source models on both ImgEdit-Bench and KRIS-Bench, while also proving to be highly competitive with several closed-source models.

| | Models | ImgEdit-Bench | GEdit-Bench | | | KRIS-Bench | | | |
|---|---|---|---|---|---|---|---|---|---|
| | | Overall | Semantic Consistency | Quality | Overall | Factual Knowledge | Conceptual Knowledge | Procedural Knowledge | Overall |
| close-source models | Gemini 2 flash (Apr. 2025) | - | 6.87 | 7.44 | 6.51 | 65.26 | 59.65 | 62.90 | 62.41 |
| | Gemini 2.5 flash (Sep. 2025) | 4.30 | 8.25 | 8.29 | 7.89 | 77.03 | 78.29 | 75.93 | 77.29 |
| | Doubao (seed edit 1.6, Apr. 2025) | - | 7.22 | 7.89 | 6.98 | 63.30 | 62.23 | 54.17 | 60.70 |
| | Doubao (Seedream 4.0, Aug. 2025) | 4.46 | 9.17 | 7.95 | 8.40 | 78.10 | 76.86 | 76.93 | 77.31 |
| | GPT4o (Apr. 2025) | - | 7.74 | 8.13 | 7.49 | 79.80 | 81.37 | 78.32 | 80.09 |
| | GPT4o (Sep. 2025) | 4.30 | 8.74 | 7.67 | 8.01 | 81.16 | 78.24 | 77.09 | 79.00 |
| open-source models | ICEdit (Zhang et al., 2025b) | 3.05 | 4.94 | 7.39 | 4.87 | 46.99 | 42.73 | 27.76 | 40.70 |
| | Omnigen (Xiao et al., 2025) | 2.96 | 5.88 | 5.87 | 5.01 | 33.11 | 28.02 | 23.89 | 28.85 |
| | Omnigen 2 (Wu et al., 2025c) | 3.44 | 7.16 | 6.77 | 6.41 | 57.36 | 44.20 | 47.79 | 49.71 |
| | BAGEL-thinking (Deng et al., 2025) | 3.56 | 7.70 | 6.51 | 6.66 | 66.18 | 61.92 | 49.02 | 60.18 |
| | BAGEL (Deng et al., 2025) | 3.20 | 7.48 | 6.80 | 6.60 | 60.26 | 55.86 | 51.69 | 56.21 |
| | Uniworld-V1 (Lin et al., 2025) | 3.26 | 4.93 | 7.43 | 4.85 | 47.71 | 44.80 | 47.92 | 50.27 |
| | Hidream-I1 (E1) (HiDream-ai, 2025) | 3.17 | 5.66 | 6.06 | 5.01 | 43.31 | 50.05 | 37.64 | 44.72 |
| | Hidream-E1.1 (Cai et al., 2025) | 3.97 | 7.15 | 6.65 | 6.42 | 43.52 | 44.71 | 36.08 | 42.25 |
| | Flux-Kontext-dev (BlackForestLabs et al., 2025) | 3.97 | 7.16 | 7.37 | 6.51 | 53.28 | 50.36 | 42.53 | 49.54 |
| | Step1X-Edit v1.1 (StepFun AI, 2025) | 3.90 | 7.66 | 7.35 | 6.97 | 53.05 | 54.34 | 44.66 | 51.59 |
| | Qwen-Image-Edit (Wu et al., 2025a) | **4.27** | 8.00 | 7.86 | **7.56** | 61.47 | 56.79 | 47.07 | 56.15 |
| | **ReasonEdit-base(Ours)** | 4.24 | 7.91 | 7.73 | 7.27 | 59.39 | 62.77 | 62.38 | 61.48 |
| | **ReasonEdit-reasoning(Ours)** | **4.27** | 7.91 | 7.80 | 7.37 | 60.77 | 65.81 | 61.62 | **63.03** |

### 4.2.1 EVALUATION ON GEDIT-BENCH AND IMGEDIT-BENCH

As shown in Table 1, our method achieves superior performance on the foundational instruction benchmarks ImgEdit-Bench and GEdit-Bench. Our method is tied for the top position on ImgEdit-Bench with a score of 4.27, and our 12B model achieves a score merely 0.19 points below that of the top-ranked open-source model, Qwen-Image-Edit (20B), on GEdit-Bench, which highlights the exceptional efficiency of our approach at only 60% of its model size.

GEdit-Bench and ImgEdit-Bench primarily evaluate a model's foundational editing capabilities. While our thinking and reflection mechanisms provide performance gains, their full impact may be less pronounced on these relatively simple tasks compared to more complex ones. This is consistent with the design of our dataset, where the thinking and reflection modules are specifically tailored for complex instructions and multi-step editing.

As shown in Fig. 3, a qualitative comparison demonstrates that our approach excels at precisely altering target areas while faithfully maintaining the integrity of unedited regions, such as backgrounds, facial features, and hairstyles. This capability addresses a key challenge in image editing by effec-

tively mitigating common failures related to consistency and fidelity, resulting in stable performance and accurate responsiveness to a wide range of commands.

### 4.2.2 EVALUATION ON KRIS-BENCH

On the KRIS-Bench, the proposed approach demonstrates the best performance among all open-source models, including those that also employ a thinking-based mechanism (e.g., BAGEL-Thinking), and surpasses several closed-source methods. The model's performance exceeds that of the 20B Qwen-Image-Edit model by 6.88 points, despite being only 60% of its size.

The performance gains are attributed to the method's ability to simplify abstract and difficult editing tasks into clear, actionable steps for the editing model. Furthermore, the reflection pipeline provides a crucial mechanism to analyze the correctness of an edit and formulate strategies for improvement. This iterative process of self-correction allows the model to identify and rectify subtle errors, effectively mitigating hallucination and improving overall fidelity. The method's demonstrated effectiveness on both complex and simple tasks ($cf$. Sec. 4.2.1) proves its versatility and robust generalization.

As shown in Fig. 3, many methods often misinterpret or fail to respond correctly to abstract or complex instructions. Our proposed thinking module effectively aids the editing model in understanding such instructions and executing them accurately. Furthermore, the reflection pipeline enhances this process by enabling the model to identify and rectify subtle errors, formulate precise refinement strategies, and prevent the compounding of mistakes that are common in multi-step editing tasks. Simultaneously, many models struggle with maintaining consistency in complex scenarios, often leading to unintended alterations in unedited regions because they lack a robust understanding of the entire scene's structure. In contrast, our approach ensures high consistency by faithfully preserving elements that should remain unchanged.

### 4.3 ABLATION STUDIES

To systematically evaluate the contribution of each component of the proposed method, a series of ablation studies are conducted on the KRIS-Bench, as its abstract and challenging nature makes it an ideal testbed for verifying the reasoning and reflection capabilities of the model. Three distinct groups of experiments are designed: first, to assess the impact of fine-tuning the base MLLM model (Qwen2.5VL 7B Instruct, hereafter referred to as Qwen); second, to investigate the individual and combined effects of the thinking and reflection modules; and third, to compare the effectiveness of the proposed single-image reflection pipeline against a single-pass dual-image approach.

Table 2: Ablation of Multi-Stage Training. This table evaluates the performance contributions of each stage in the training pipeline, from the pre-trained baseline to the final unified model, highlighting the cumulative benefits of fine-tuning the generator and reasoning modules at each step.

| Methods | KRIS-Bench | | | |
| --- | --- | --- | --- | --- |
| | Factual Knowledge | Conceptual Knowledge | Procedural Knowledge | Overall |
| Pre-trained Generator (Step1X-Edit V1.1) | 53.05 | 54.34 | 44.66 | 51.59 |
| Pre-trained Generator + Qwen Reasoning | 54.05 | 57.44 | 41.26 | 52.41 |
| Pre-trained Generator + Qwen-tuned Reasoning | 55.34 | 62.06 | 45.24 | 55.70 |
| Base Generator W/O Reasoning | 55.80 | 55.28 | 43.78 | 52.74 |
| Base Generator + Qwen-tuned Reasoning | 55.13 | 62.11 | 46.51 | 55.94 |
| **Unified Tuned (Ours)** | 60.77 | 65.81 | 61.62 | **63.03** |

**Impact of Multi-Stage Training.** To evaluate the specific contribution of the reasoning learning stage, we compare the performance of the Pre-trained Generator when integrated with either a base (untuned) Qwen model or a fine-tuned Qwen model. When the Pre-trained Generator is augmented with the base Qwen model leveraging our thinking and reflection mechanism, only a marginal performance gain of 0.82 points is observed. In contrast, fine-tuning Qwen on our reasoning data consistently and significantly outperforms this base configuration. This highlights that foundational multimodal large language models, without domain-specific adaptation, struggle to effectively grasp the nuances of image editing, thereby underscoring the critical necessity of tailoring the MLLM to these specific demands. After the edit learning stage, in isolation, the Base Generator achieves a degree of performance improvement over the Pre-trained Generator, demonstrating its role in adapting the generative capabilities. Finally, this multi-stage strategy culminates in the optimal performance

of the unified training, providing a substantial performance increase from the Base Generator + Qwen-tuned Reasoning model to the Unified Tuned model (55.94 vs. 63.03), validating the synergistic benefits of training the entire pipeline as a whole.

Table 3: Ablation Study on the Contributions of the Thinking and Reflection Modules. The table shows the performance of four model variants on the KRIS-Bench, demonstrating the benefits of each component and the synergy of their combination.

| Methods | KRIS-Bench | | | |
|---|---|---|---|---|
| | Factual Knowledge | Conceptual Knowledge | Procedural Knowledge | Overall |
| Unified-tuned Generator | 59.39 | 62.77 | 62.38 | 61.48 |
| Unified-tuned Generator + thinking | 60.60 | 65.90 | 60.87 | 62.83 |
| Unified-tuned Generator + reflection | 60.40 | 62.46 | 62.58 | 61.76 |
| **Unified-tuned Generator + thinking + reflection** | 60.77 | 65.81 | 61.62 | **63.03** |

**Ablation of Thinking and Reflection.** To understand the individual and combined contributions of the thinking and reflection modules, four variants are compared: (1) a baseline model without either module; (2) a model with only the thinking module; (3) a model with only the reflection module; and (4) the full model incorporating both. The results on KRIS-Bench (see Tab. 3) show a gradual improvement in performance with the addition of each component. The thinking module alone provides a significant performance boost, confirming its effectiveness in handling complex instructions. The thinking + reflection module proves beneficial on both GEdit-Bench (see Tab. 1) and KRIS-Bench, as it effectively rectifies errors. The full model, with both modules integrated, achieves the highest scores, highlighting the synergistic relationship between understanding an instruction and correcting subsequent errors.

Table 4: Ablation Study on Reflection Pipelines. The table compares the performance of three different reflection mechanisms on KRIS-Bench, highlighting the effectiveness of the proposed multi-round pipeline.

| Methods | KRIS-Bench | | | |
|---|---|---|---|---|
| | Factual Knowledge | Conceptual Knowledge | Procedural Knowledge | Overall |
| Base Generator | 55.80 | 55.28 | 43.78 | 52.74 |
| Base Generator + dual-image pipeline | 52.97 | 61.84 | 41.12 | 53.79 |
| Base Generator + single-image pipeline | 54.81 | 56.92 | 43.70 | 53.04 |
| **Base Generator + our multi-round pipeline** | 55.13 | 62.11 | 46.51 | **55.94** |

**Comparison of Reflection Pipelines.** To ensure consistency in the DiT parameters, this ablation study is conducted by combining the Base Generator (the DiT after the edit learning stage) with each of the reflection pipelines. Tab. 4 compares three distinct approaches to the reflection process—a dual-image pipeline, a pure single-image pipeline, and the proposed multi-round prior pipeline (see Sec. A.1). The dual-image pipeline, which relies on a direct comparison between the initial input and the generated output, is often prone to hallucinations. Conversely, a pure single-image approach struggles with tasks that require a clear before-and-after comparison, such as Portrait Beautification or motion/expression-related edits. As shown in the table, the proposed multi-round single-image prior pipeline is superior. This is attributed to the method's ability to combine the benefits of both approaches, allowing it to perform a self-correction loop on the generated image itself while leveraging key prior information from the multi-round process.

## 5 CONCLUSION

In this work, we present ReasonEdit, a fundamental image editing framework that demonstrates the crucial role of explicit reasoning in achieving robust and versatile performance. The proposed method introduces a novel pipeline with two core capabilities: thinking and reflection. By training these capabilities on a curated collection of Thinking Pairs and Reflection Triples, the framework learns to convert abstract user requests into actionable commands and to perform self-correction in an iterative loop. Extensive experiments on a range of benchmarks validate the efficacy of this approach, with the model achieving state-of-the-art performance among open-source methods on ImgEdit and Kris-Bench while remaining highly competitive with several closed-source models. This work provides a new perspective on reasoning-enhanced image editing, showing that a structured pipeline for instruction understanding and self-correction is vital for building models that can handle both simple and complex editing tasks with high fidelity and consistency.

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

# A APPENDIX

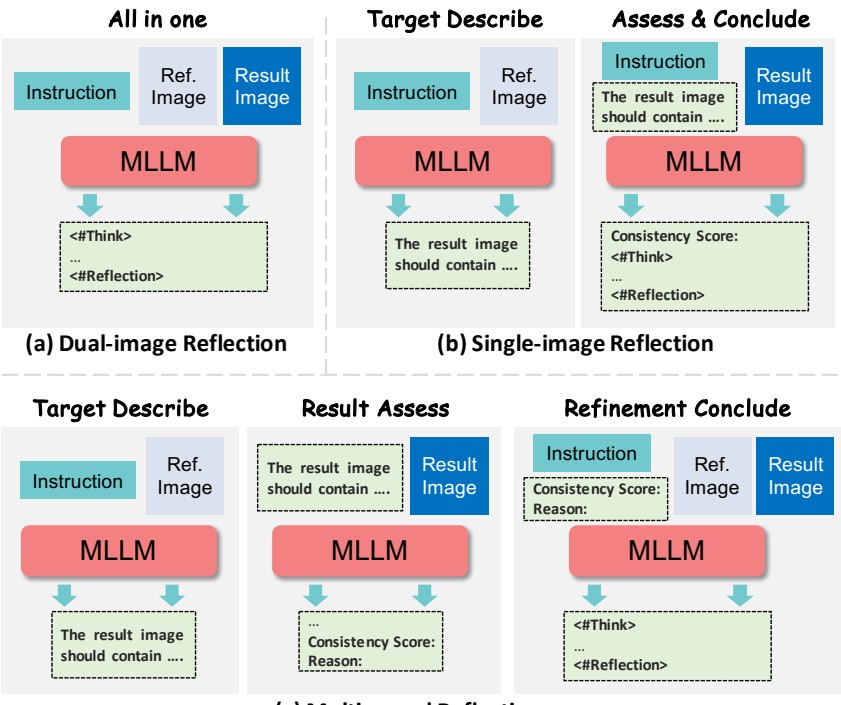

Figure 4: Three distinct MLLM-based reflection pipelines for image editing. (a) **Dual-image Reflection** processes the instruction, reference image, and result image simultaneously in a single MLLM call to produce a combined thinking and reflection output. (b) **Single-image Reflection** decomposes this into two sequential MLLM calls: first, generating a target description from the instruction and reference image, and then assessing the result image against this description to provide a consistency score and reflection. (c) Our proposed **Multi-round Reflection** pipeline further refines the process into three dedicated stages: (1) *Target Describe*, which formulates a target image description from the input instruction and reference image; (2) *Result Assess*, which evaluates the generated image against this target description to output consistency scores and reasons; and (3) *Refinement Conclude*, where the MLLM analyzes the assessment to provide success/failure judgments and, if necessary, detailed instructions for further image modification, leveraging both the reference and result images. This multi-round approach enables a more granular and iterative refinement of image editing outcomes.

## A.1 ILLUSTRATION OF REFLECTION PIPELINES

We conduct an ablation study on the proposed multi-round reflection pipeline against two alternative designs, as presented in Tab. 4 and further illustrated in Fig. 7. The Dual Image Reflection pipeline directly inputs the reference image, edit instruction, and result image, tasking the MLLM with generating thinking and reflection concurrently. Nevertheless, we found that current MLLMs frequently exhibit hallucinations in image editing tasks under this unified input scheme. Our investigation then led to the Single Image Reflection pipeline, which first guides the MLLM to describe the target image based on the edit instruction and reference image. Subsequently, using this target description, the MLLM evaluates the result image, offering detailed reasoning, identifying failures, suggesting refinements, or confirming success. A key drawback here is that the MLLM loses the essential context of the reference image during its final conclusion, leading to less effective assessments. The proposed pipeline addresses these limitations by decomposing the reflection into three distinct sub-procedures: target description, result assessment, and refinement conclusion. In the initial two sub-procedures, the MLLM receives only a single image as input (e.g., reference image for

target description, result image for assessment), which significantly enhances accuracy. In the final stage, the MLLM is provided with all relevant information to formulate a comprehensive conclusion for subsequent actions, thereby maintaining full contextual awareness.

# B MORE RESULTS

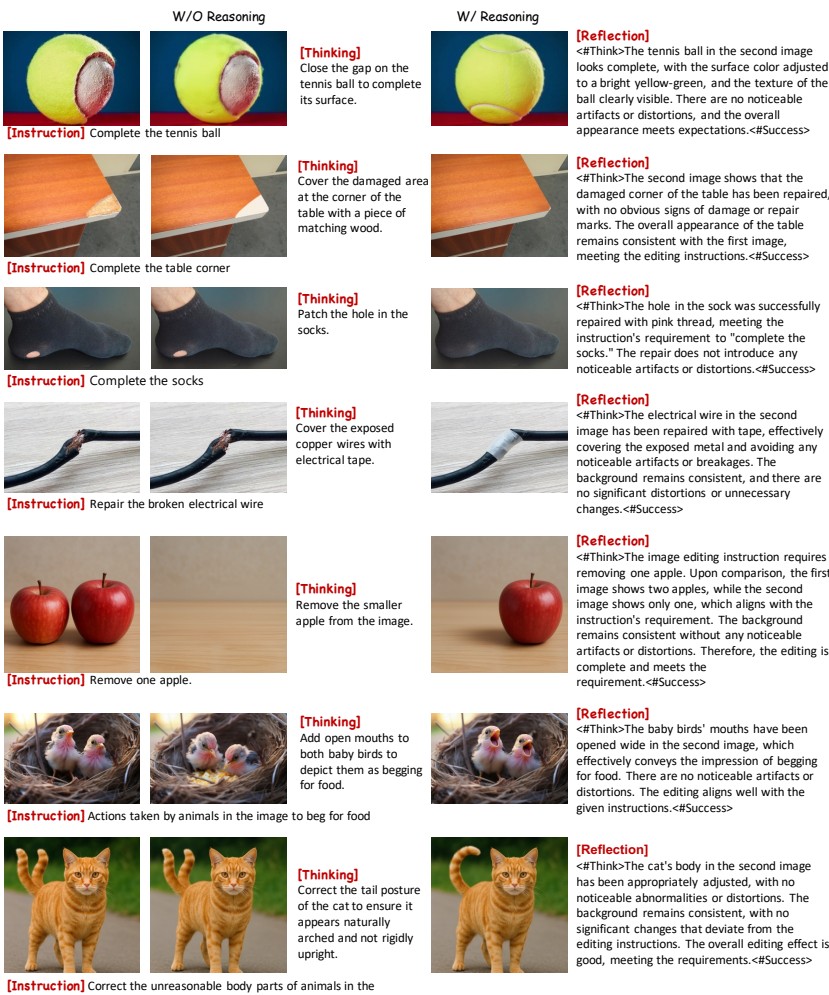

Figure 5: Qualitative examples illustrating the thinking capabilities of our MLLM Reasoner across various image editing instructions.

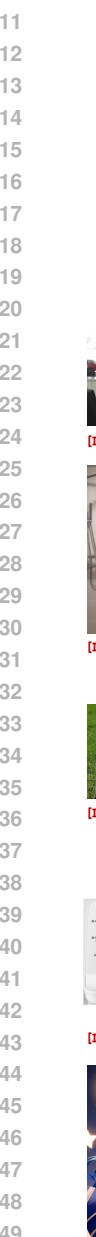

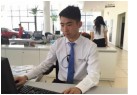
**[Think]** Apply a background blur effect to the entire scene.

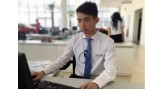
**[Reflection Round 1]**
<#Think>The background blur effect in the second image is not very noticeable, with some details still visible. Further adjustments are needed to enhance the blur effect.<#Reflection>**Add more blur to the background to make the details less distinguishable.**

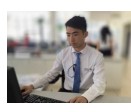
**[Reflection Round 2]**
Add more blur to the background to make the details less distinguishable. <#Think>The background blur effect in the second image has been successfully applied, meeting the instruction requirements. There are no obvious artifacts or distortions, and the overall background appears naturally blurred.<#Success>

**[Instruction]** Add a background blur filter.

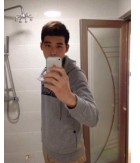

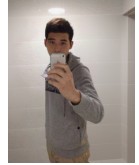
**[Reflection Round 1]**
<#Think>The background in the second image has been simplified but still contains elements of the original background. The primary background objects such as the shower and door frame have been removed, but the wall tiles and ceiling details remain visible. To fully meet the instruction's requirement to remove the background, these remaining details need to be completely obscured. <#Reflection>**To fully remove the background, all remaining elements such as the wall tiles and ceiling need to be obscured.**

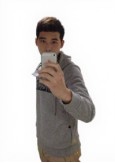
**[Reflection Round 2]**
<#Think>The background has been successfully removed, leaving only a pure white background. The person's clothing and posture have remained consistent, with no noticeable distortion or artifacts.<#Success>

**[Think]** Remove the bathroom background (shower, door, mirror).

**[Instruction]** Remove the background for me.

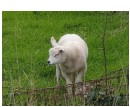

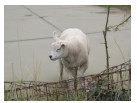
**[Reflection Round 1]**
<#Think>The background has been changed from green grass to a concrete surface, which meets the instruction requirements. However, some vegetation on the edge of the concrete is still visible, which is an artifact that does not align with the instruction's requirement for a completely solid concrete surface. Therefore, further modification is needed. <#Reflection>**Secondary editing instruction: Remove the remaining vegetation from the edges of the concrete surface.**

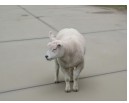
**[Reflection Round 2]**
<#Think>The background has been successfully adjusted to a concrete ground, with no obvious artifacts or distortions. Other areas have not been altered, meeting the editing instruction requirements.<#Success>

**[Think]** Replace the grassy field background with concrete ground.

**[Instruction]** Adjust the background to a concrete ground.

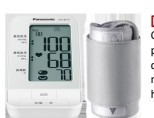

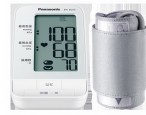
**[Reflection Round 1]**
Change the blood pressure values displayed on the monitor to indicate hypertension. <#Think>The second image shows a blood pressure meter displaying a reading of 100/68 mmHg, which does not meet the requirement of changing the blood pressure value. There are no significant artifacts or distortions, and the overall editing is acceptable. <#Reflection>**Change the blood pressure reading displayed on the meter to 170/90 mmHg.**

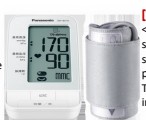
**[Reflection Round 2]**
<#Think>The blood pressure monitor in the second image has been edited to display a systolic pressure of 170 mmHg and a diastolic pressure of 90 mmHg, indicating hypertension. This aligns with the given editing instruction.<#Success>

**[Think]** Change the blood pressure values displayed on the monitor to indicate hypertension.

**[Instruction]** Change to hypertension values

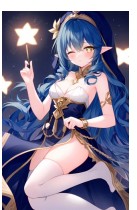

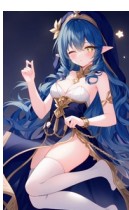
**[Reflection Round 1]**
Remove the magic wand and stars from the background. <#Think>The second image has the magic wand removed, but the stars have not been deleted. This does not fully comply with the instruction to remove both the magic wand and the stars.<#Reflection>**Please delete all the stars in the image.**

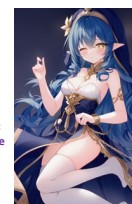
**[Reflection Round 2]**
<#Think>The magic wand and stars have been successfully removed, meeting the editing instructions.<#Success>

**[Think]** Remove the magic wand and stars from the background.

**[Instruction]** Remove the magic wand and stars from the background.

Figure 6: Qualitative multi-round examples illustrating the thinking and the reflection progressively correct and improve the quality of the generate results.

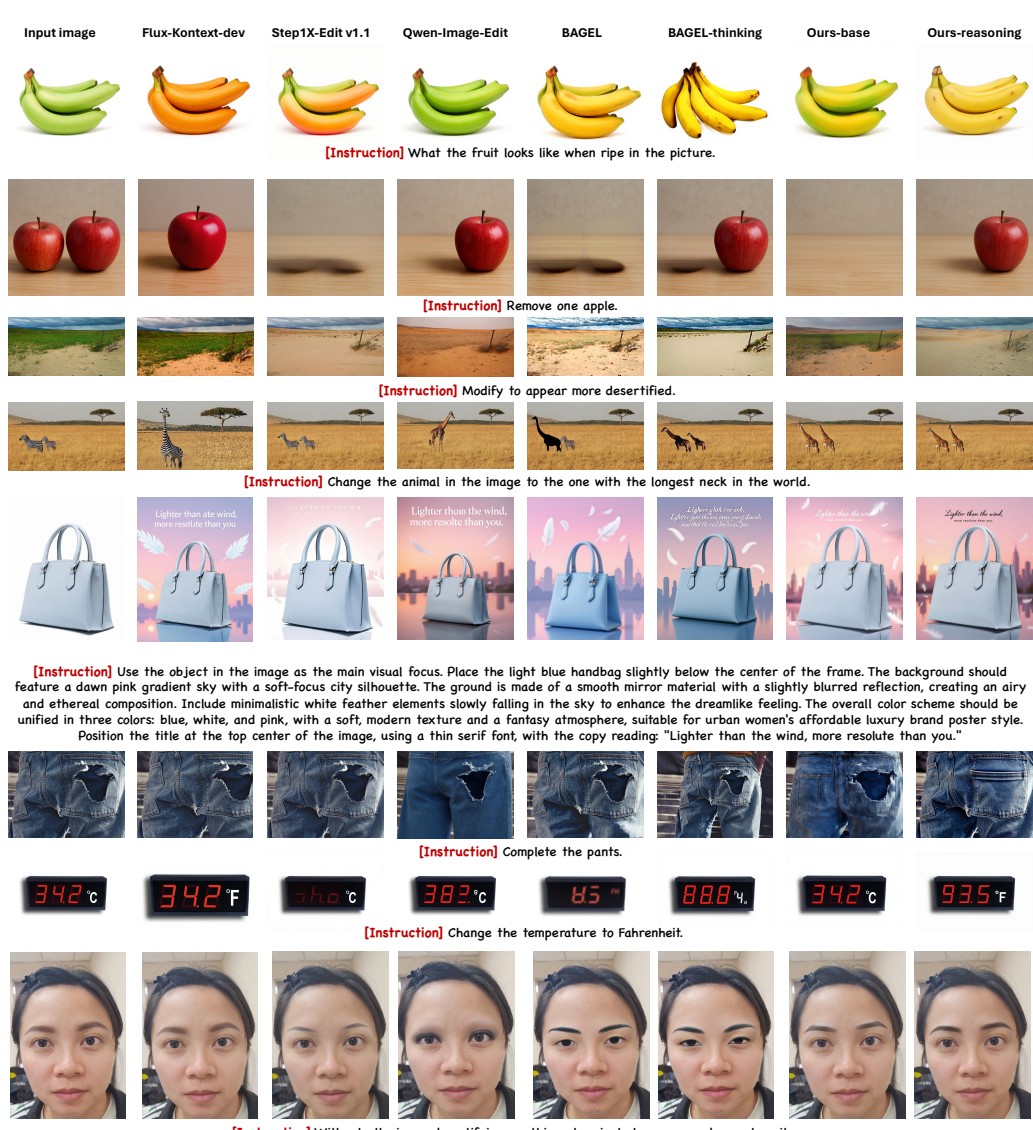

Figure 7: More qualitative comparison of our method and state-of-the-art approaches.

