# OpenReview forum: "ReasonEdit: Towards Reasoning-Enhanced Image Editing Models"
_ICLR.cc/2026/Conference — ICLR 2026 Conference Withdrawn Submission_

### Official Review · Reviewer_SKtX · 2025-10-26

**Soundness:** 3
**Presentation:** 2
**Contribution:** 2
**Rating:** 6
**Confidence:** 3

**Summary:**

This paper targets reasoning-enhanced image editing models. On the basis of the most commonly used framework: multi-model large language model + diffusion model, where the former one is frozen, it introduces a new thinking–editing–reflection loop. Thinking means use MLLM to understand the instruction while reflection means an iterative self-correction. The whole model can trained in an end-to-end manner. Extensive experiment results prove the effectiveness of the proposed method.

**Strengths:**

1. Overall this paper is well-organized and easy to read. The figures illustrates the idea, especially the thinking–editing–reflection loop very clearly.
2. The experiment section is extensive. It conducts experiments on mainstream benchmarks, and provide comparison results with enough baseline method. So I think the experiment results can be convincing.
3. The idea is simple yet reasonable, it makes the image-editing system more consistent with human being. I can understand that this RL-like framework is effectiveness.

**Weaknesses:**

1. The idea is reasonable, but the novelty is just ok, but not high. I do not mean 'pursuing novelty' here, but I am happy to see more discussions of novelty or any other interesting parts in model design.
2. The authors can have some discussions on the limitation of the proposed method, including some failure case. I think it will be beneficial to the community.
3. I also want to see some discussions on the training cost or the stability of the model.
4. I think the introduction part of the paper can be further polished by adding some more descriptions about model design and performance.

**Questions:**

See weaknesses.

---

### Official Review · Reviewer_D8fU · 2025-10-27

**Soundness:** 3
**Presentation:** 3
**Contribution:** 3
**Rating:** 4
**Confidence:** 4

**Summary:**

This paper introduces ReasonEdit, a reasoning-augmented framework for image editing that integrates a multimodal large language model (MLLM) as a Reasoner and a diffusion transformer (DiT) as a Generator. The framework builds a closed loop of thinking–editing–reflection to improve reasoning depth, instruction following, and iterative refinement.

**Strengths:**

- Well-motivated and clearly structured framework: The motivation—to address the limited reasoning capability of existing MLLM-frozen editing pipelines—is clearly articulated. The decomposition into a Reasoner (MLLM) and Generator (DiT), coupled through thinking and reflection cycles, provides a coherent design that is easy to follow and conceptually elegant.
- Innovative data construction tailored for reasoning-aware editing: The paper goes beyond standard instruction-image datasets by introducing Thinking Pairs and Reflection Triples. These datasets explicitly encode step-by-step reasoning and self-evaluation, offering a structured way to teach “how to think and correct.”
- Systematic comparison of reflection pipelines: The paper compares three reflection strategies and reports consistent superiority of the latter, providing useful insight for future research.

**Weaknesses:**

- Dataset generation and reproducibility insufficiently detailed: The Thinking and Reflection datasets rely heavily on automated labeling with advanced MLLMs, but the paper does not disclose which annotators or models were used, nor any quality control metrics (agreement rate, filtering thresholds, or bias mitigation). Without these details, reproducibility and data reliability remain unclear.
- Evaluation over-relies on GPT-4/4o automatic scoring: Most benchmarks use GPT-based metrics (VIEScore, GPT-4o evaluation). While common, this raises the concern of co-training bias. The absence of human A/B testing or inter-rater agreement weakens claims of perceptual superiority. Cross-evaluator consistency would strengthen the argument.
- Fairness and statistical significance not established: Comparisons in Table 1 include closed-source models under heterogeneous inference budgets (temperature, reasoning steps, reflection rounds). No standard deviation or significance test is reported. Multiple runs with fixed budgets are needed to support SOTA claims.
- Marginal improvement on basic editing tasks lacks deeper analysis: The paper notes limited gain for low-level edits but does not provide per-category error breakdowns (e.g., color, geometry, style). An analysis of failure cases would clarify when reasoning adds value versus when it is redundant.
- Computation and latency overhead not quantified: Reasoning and reflection introduce extra forward passes. Although training GPU counts are reported, inference-time cost, average reflection rounds, and latency–quality trade-offs are not provided. This information is critical for practical deployment.
- Data scale and domain sensitivity analysis missing: The ratio of T2I vs editing samples is listed but no study of how dataset composition affects performance is included. A sensitivity or ablation study would make the dataset design more instructive for replication.

**Questions:**

- How many reasoning/reflective steps are used during inference? What is the latency–accuracy trade-off curve?
- Have human or cross-model (non-GPT) evaluations been performed to confirm robustness on KRIS-Bench?

---

### Official Review · Reviewer_qgb8 · 2025-10-28

**Soundness:** 3
**Presentation:** 3
**Contribution:** 3
**Rating:** 4
**Confidence:** 4

**Summary:**

This paper introduces ReasonEdit, a reasoning-enhanced image editing framework that integrates explicit “thinking” and “reflection” modules within an MLLM-Diffusion model architecture. The thinking module transforms abstract, ambiguous, or colloquial editing instructions into clear, actionable commands, while the reflection module iteratively reviews and revises intermediate edits to improve accuracy. The system is trained in a multi-stage pipeline on curated “Thinking Pairs” (abstract-to-concrete instructions) and “Reflection Triples” (input, intermediate, and target images with reflections/diagnostics). Experiments on three benchmarks (GEdit-Bench, ImgEdit-Bench, and KRIS-Bench) show that ReasonEdit achieves significant gains over baseline and related methods.

**Strengths:**

- The introduction of explicit, modular “thinking” (instruction grounding/decomposition) and “reflection” (iterative self-correction) mechanisms directly within an image editing pipeline is well-motivated and appropriately positioned with respect to recent advances in reasoning for multimodal models.
- The work details a robust data pipeline (including the construction of both “Thinking Pairs” and “Reflection Triples”) to support supervised training for both the reasoning and editing aspects. The scale and systematic curation add significant credibility.
- Quantitative results demonstrate competitive performance on open-source benchmarks, with strong gains on the challenging KRIS-Bench reasoning tasks. The improvement is backed by ablation studies which isolate the effects of reasoning modules and training stages.

**Weaknesses:**

- While the paper is methodologically sound, it does not provide deeper theoretical or formal justification for why incorporating reasoning delivers better generalization or robustness beyond anecdotal/empirical evidence. There is no formal analysis of potential failure modes, e.g., when thinking/reflection might increase hallucination or overfit to annotation artifacts.
- The work focuses solely on the image editing scenario and largely benchmarks against datasets that were partially constructed or curated by the authors' proposed pipeline. There is little evidence that the discovered benefits in “reasoning” generalize to other vision or multimodal tasks, or hold when adversarial or out-of-distribution manipulations are presented. Experiments on more diverse, external, or “in-the-wild” benchmarks would increase confidence.
- Although the dataset construction process is outlined in Section 3.1, there is insufficient detail about the annotation guidelines, inter-annotator agreement, or concrete measures to ensure that the paired data do not contain distributional or conceptual shortcuts. It is unclear how robust the datasets are to annotation artifacts or if “reflection triples” can introduce biases through automated or manual curation.
- As highlighted in the ablation studies and Figure 4, the multi-round reflection pipeline is more complex and computationally intensive than simpler dual- or single-image reflection baselines. However, the paper does not provide computational cost, inference speed, or training efficiency trade-offs, nor does it discuss scalability limits for practical real-world deployments.
- While the presentation of loss functions is generally correct, the flow matching formulation could be clearer regarding how the conditioning $c$ (for “thinking”/“reflection” outputs) is incorporated at each stage of the generator pipeline, and in how gradients flow in the combined joint loss $\mathcal{L}_{\text{joint}}$ during optimization. The paper also glosses over whether the LoRA adaptation (for reasoning module) causes optimization instabilities or representation collapse when coupled with diffusion-based training.
- The qualitative figures mainly highlight positive examples. There is limited discussion about concrete cases where the reasoning modules produced spurious interpretations, incorrect refinements, or exacerbated model failures that would be helpful for practitioners and for future research.
- While some mention is made of model size (e.g., ReasonEdit 12B vs. Qwen-Image-Edit 20B), detailed analysis on the effect of reasoning on memory/compute overhead, scalability advantages, or resource constraints (such as using smaller/frozen models) is missing. This limits insight into deployment contexts.

**Questions:**

- Can the authors provide empirical numbers on the additional computational and wall-clock costs for the multi-round reflection pipeline compared to single/double-pass alternatives? Is the reflection pipeline amortizable at inference in practical settings?
- What are unambiguous scenarios or instruction types where reasoning/reflection actually induces overfitting, hallucination, or systematic errors? Are there examples where “thinking” decompositions or “reflection” cycles degrade performance?
- Could the authors share more about annotation protocols and provide quantitative measures of instruction clarity, reflection reliability, or inter-annotator consistency for their curated datasets?
- How does the framework fare when presented with entirely new image domains, unseen instruction styles, or non-editing vision-language reasoning tasks? Would a similar paradigm generalize?
- Is the impressive efficiency of ReasonEdit at 60% model size sustained across smaller or larger variants? Is performance stable for lower-resource/frozen MLLMs or smaller DiTs?

---

### Official Review · Reviewer_rfEs · 2025-11-01

**Soundness:** 3
**Presentation:** 3
**Contribution:** 3
**Rating:** 4
**Confidence:** 4

**Summary:**

The authors propose ReasonEdit, a framework that integrates the reasoning capabilities of multimodal large language models (MLLMs) into diffusion models to enable complex instruction-based image editing. The authors introduce thinking–editing–reflection workflow and construct the corresponding dataset to fine-tune the unified model for iterative, reasoning-driven refinement of editing results. Extensive experiments show strong editing performance on ImgEdit, GEdit-Bench and Kris-Bench.

**Strengths:**

1. Clear Motivation. ReasonEdit effectively leverages the reasoning capabilities of MLLM to enhance image editing performance. The authors explore both thinking and reflection modes for an instruction-based editing task. Instead of treating the MLLM as a frozen feature extractor, the authors jointly optimize MLLM with the diffusion decoder based on their reasoning-enhanced dataset, improving the performance under abstract instructions.
2. Well-designed data curation and training strategy. The proposed dataset is thoughtfully constructed, comprising 1) the thinking pairs consist of abstract-to-concrete instruction pairs, and 2) the reflection triples, which consist of an input Image, a generated image, and a target Image. Their training strategy is well-motivated and consists of three stages：1) fine-tune MLLM while freezing the diffusion decoder to learn reasoning, 2) fine-tune DiT for editing learning, and 3) perform joint fine-tuning for unified optimization.
3. Comprehensive evaluation. Comprehensive experiments and ablation studies on ImgEdit-Bench, GEdit-Bench and Kris-Bench demonstrate the model’s strong capability for abstract, instruction-based image editing

**Weaknesses:**

1. The "reflection" mechanism is designed as an "iterative self-correction and optimization" process. This iterative process inevitably increases inference time (Latency) and computational overhead, making it slower than single-pass editing models. However, the paper does not report evaluations on inference speed, computational cost, or the average number of reflection rounds required for success throughout. This is a significant limitation for the practical application of the model.
2. Although the model performs excellently on KRIS-Bench, which is designed for high-difficulty reasoning tasks, its improvement on standard basic editing benchmarks is not significant. On GEdit-Bench, the score of Ours-reasoning is lower than that of Qwen-Image-Edit, which is also an open-source model.

**Questions:**

1. How to define the upper bound (or lower bound) editing time of this method?
2. Should it report the model size and the running efficiency of each method?
3. In experiments, this paper only reports the overall quality on the whole benchmark; it would be better to show the editing performance in different aspects since image editing contains many different tasks.

---

### Note · Authors · 2025-11-13

I have read and agree with the venue's withdrawal policy on behalf of myself and my co-authors.